# Fear God, Not COVID-19: Is Conservative Protestantism Associated with Risky Pandemic Lifestyles?

**DOI:** 10.3390/healthcare11040582

**Published:** 2023-02-15

**Authors:** John P. Bartkowski, Katherine Klee, Terrence D. Hill, Ginny Garcia-Alexander, Christopher G. Ellison, Amy M. Burdette

**Affiliations:** 1Department of Sociology, The University of Texas at San Antonio, San Antonio, TX 78249, USA; 2Department of Sociology, Florida State University, Tallahassee, FL 32306, USA

**Keywords:** religion, faith, conservative Protestant, evangelical, fundamentalist, health, disease, illness, coronavirus, COVID-19, pandemic, vaccine, lifestyles

## Abstract

Previous research has established attitudinal and behavioral health variations in relation to the COVID-19 pandemic, but scholarship on the religious antecedents associated with these outcomes has only recently gained momentum. Rhetoric from some leading conservative Protestants in the U.S. has underplayed the threat of the pandemic and may have contributed to unhealthy pandemic behaviors within this faith tradition. Moreover, previous inquiries have revealed that conservative Protestantism’s otherworldly focus can thwart personal and community health. We use nationally representative data to test the hypotheses that, compared with other religious groups and the non-religious, conservative Protestants will tend to (1) perceive the pandemic as less threatening and (2) engage in riskier pandemic lifestyles. These hypotheses are generally supported net of confounding factors. We conclude that affiliation with a conservative Protestant denomination can undermine public health among this faith tradition’s adherents and may therefore compromise general health and well-being during a pandemic. We discuss the implications of these findings, offer recommendations for pandemic health promotion among conservative Protestants, and delineate promising avenues for future research on this important topic.

## 1. Introduction

American religious and political conservatism have a long history of alliance, a bond seemingly further cemented by the COVID-19 pandemic. As just one case among many, Tate Reeves, the Republican Governor of Mississippi, attracted national media attention during the pandemic by stating that the preponderance of highly religious people in his state value eternal life over worldly concerns and are therefore less prone to fearing COVID-19. Said Reeves, “I am often asked by some of my friends on the other side of the aisle about COVID … and why does it seem like folks in Mississippi and maybe in the Mid-South are a little less scared [of the pandemic], shall we say … When you believe in eternal life—when you believe that living on this earth is but a blip on the screen, then you do not have to be so scared of things” (as quoted in *Newsweek*) [1]. Reeves was careful to qualify these remarks by stating that people should also take necessary precautions. However, such qualifications were mingled with a relatively restrained Republican approach in the South and around the nation to scientifically supported public health mitigation measures (e.g., mask mandates, aggressive vaccine promotion, and vaccine requirements for healthcare workers). Quite notably, Mississippi was among the states in the U.S. with the slowest vaccine uptake and pronounced fallout from the pandemic, sometimes adopting approaches contrary to the advice of its chief medical officer [2].

Populated heavily by white conservative Protestants (evangelicals, fundamentalists, and Pentecostals) who are concentrated in areas with poor public health and elevated premature mortality rates [3], Mississippi is widely considered the “buckle” of the Bible Belt that is, essentially, the American South. Yet, such sentiments are probably not idiosyncratic to Mississippi or the largely conservative Protestant American South. There is a set of unique cultural values that tie together conservative Protestants across the nation. Some of these distinctive views were leveraged into widespread skepticism and, at times, outright disinformation expressed by nationally renowned conservative Christians (e.g., Eric Metaxas) about COVID-19 and vaccinations [4]. During the onset of the pandemic, the clash of values with church-gathering restrictions reached a fever pitch through worship protests, that is, Christian praise song concerts and “prayer sessions” that intentionally flouted health regulations designed to promote social distancing and reduce disease transmission at large public events [5,6]. Many of these protest prayer sessions were led by Sean Feucht, a Californian music missionary turned defeated Republican congressional candidate who had the support of several conservative religious and political leaders (e.g., Josh Hawley, Mike Huckabee, former Vice President Mike Pence, and former president Donald Trump). In the wake of Black Lives Matter protests and COVID-19-specific congregational lockdowns, Feucht targeted protest sites (e.g., the George Floyd memorial in Minneapolis, Capitol Hill Occupied Protest [CHOP] in Seattle, and mobilization in downtown Portland) where he promised to transform these locations from “riots to revivals” through prayer and musical worship [6]. These worship protests, singing “all about Jesus” with intentional disregard for pandemic safety measures, gathered support largely from thousands of fundamentalist Christians and some conservative Roman Catholics who felt their religious freedom was “under attack” by state-mandated closures of houses of worship [5]. Feucht soon thereafter created a new movement, named Hold the Line, seeking to “provoke believers across the nation to take a stand” for conservative religious and political values against “cancel culture” [6].

Vocal opposition to pandemic safety measures expressed by conservative Protestant leaders was given extensive media coverage and may have influenced grassroots perspectives on public health during the pandemic. But the relationship between elite rhetoric and public opinion on this issue proved to be complicated. Religious distrust of medical science preceded the pandemic. Individuals, especially those who believe in the ability of God or a higher power to intervene in the world, have a long history of vaccine hesitancy and mistrust of health science [7]. Yet, as worship gathering restrictions were initially announced in the early days of COVID-19 transmission, they received broad public support. In a poll released in May 2020, a representative survey of U.S. adults revealed that two thirds of Americans (including evangelical Christians and Republicans) did not believe that “prohibiting in-person religious services” during the pandemic was a violation of religious freedom [8]. In fact, less than 9% believed in-person religious services should be permitted without pandemic restrictions. Consequently, perceptions about the religious contours of pandemic health exhibited by elite conservative religious critics seemed to differ from the views of average Americans and congregants in the early stages of the pandemic. In some respects, American religion appeared to gain momentum as the pandemic took hold. Early pandemic data suggest that some Americans who were relatively disconnected from faith communities prior to the initial lockdown reported attending in-person worship gatherings more frequently during late April or early May 2020 [9]. It was these Americans who often tested positive for COVID-19 by August 2020, which suggests that their reported in-person worship attendance at the height of lockdown restrictions may not have reflected their stronger commitment to religious communities. Rather, their very attendance may have been a protest action against pandemic restrictions or an effort to ameliorate feelings of social isolation. Religious gatherings could be seen as offering a reprieve from social isolation and have often promoted hope as well as an end to mental, physical, and spiritual suffering. In short, congregations may have been a pandemic shelter for even the irreligious. For their part, many congregational leaders across faith communities encouraged their congregants to receive the COVID-19 vaccine once available and to respect health and safety measures (e.g., masking, social distancing, lockdown measures) for the sake of individual and community health [10]. In January of 2021, a survey of evangelical leaders found that 95% supported receiving the COVID-19 vaccination and 89% would encourage others to do the same [11]. However, faith leaders risked losing influence over their congregants as mainstream conservative media outlets and political leaders became more vocal about their mistrust of coronavirus vaccines and pandemic safety precautions [10].

Given such messaging and mobilization among conservative religious elites, is it reasonable to suspect that grassroots conservative Protestants are more inclined toward COVID-19 skepticism than their religious peers? It seems likely that affiliation with a faith tradition that prioritizes otherworldly and individualistic convictions can relativize public health considerations in a manner that could undermine pandemic threat perceptions and protective behaviors. Moreover, the epistemology of conservative Protestantism emphasizes scriptural knowledge over scientific discovery. Previous research has underscored anti-intellectual tendencies that can undermine both verbal acumen and scientific literacy among conservative Protestants [12,13], and these dispositions could contribute to COVID-19 skepticism. An examination of empirical data is needed to test hypotheses about conservative Protestant pandemic perceptions and practices and, if such suspicions are confirmed, to engage and even challenge religious barriers to public health promotion during incidences of high vulnerability to life-threatening diseases. Recognizing the nature, scope, and antecedents of health science opposition, religious or otherwise, is a critical consideration for public health researchers.

Our study builds on previous scholarship, which is reviewed in the following section, that has revealed religious variations in COVID-19 response. We also review and extend earlier scholarship that has underscored how conservative Protestantism’s otherworldly focus can thwart community health. We generate hypotheses from previous research and anticipate that, compared with other religious groups and the non-religious, conservative Protestants will tend to (1) perceive the pandemic as less threatening and (2) engage in riskier pandemic lifestyles. We use nationally representative data to test these hypotheses, which are generally supported. Affiliation with a conservative Protestant denomination can undermine public health among this faith tradition’s adherents. During a pandemic or high-transmission periods of infectious diseases, such views may also compromise general health and well-being. These findings have important implications, including the adoption of culturally appropriate strategies for pandemic health promotion in conservative Protestant communities. We are careful to call attention to the limitations of our study while identifying promising avenues for future research.

### Empirical and Theoretical Background

American religion has been shown to influence health outcomes in complex ways [3,14,15]. Some of this complexity emanates from the diverse expressions of faith in the United States. For example, communities with high concentrations of Roman Catholics exhibit lower rates of infant and adult mortality, even as these rates are significantly elevated in largely conservative Protestant locales [3,15,16]. Prior research has emphasized how the otherworldly and individualistic values of conservative Protestantism could undermine the prioritization of public health considerations and detract from a collective investment in a robust healthcare infrastructure. As a more this-worldly and collectivist faith, Catholicism is an intriguing counterpoint to conservative Protestantism.

Research on religious variations in pandemic responses has emerged more recently. According to Beyerlein and colleagues, the dominant religious response to COVID-19 was faith maintenance, as 70% of U.S. believers declared that the coronavirus outbreak did not influence their faith and some even found that their faith strengthened [17]. Noted were two theodicy-informed interpretations of how the COVID-19 pandemic strengthened the faith of many Americans. The first was interpreting the coronavirus outbreak as God’s way of telling humanity to change the way we live, and the other was believing that God will act as a personal shield against the virus. Data from a national probability sample of the U.S. population revealed that highly religious individuals, namely evangelicals, suffered less distress in March 2020 [18]. These individuals were less likely to see the COVID-19 outbreak as a crisis and were less inclined to support public health measures to limit the spread of the virus. Religion can, undoubtedly, offer comfort and strength during times of crisis and, in the case of pandemic conditions, protect the mental health of the devout and formerly irreligious who have since turned to faith [18].

In a pathbreaking ecological investigation, DeFranza and colleagues utilized shelter-in-place directives issued in early 2020 as an intervention for their quasi-experimental study using data from select populous metropolitan areas in the United States [19]. They examined variations in directive adherence over 30 days as a function of community-level religious composition. Metropolitan communities with higher concentrations of religious persons exhibited lower adherence to shelter-in-place directives, thereby providing evidence of religious resistance to these directives. The researchers surmised that such directives were perceived as an infringement on religious freedom. The alternative hypothesis that religion may promote greater rule-abiding and altruistic behavior, thereby leading to greater compliance with shelter-in-place directives, was not supported. Another line of inquiry, focused instead on personal religiosity, revealed that a belief in an engaged God (i.e., loving, involved, caretaking) was correlated with greater mistrust of COVID-19 vaccines, an association that was amplified for Hispanic and less educated Americans [20]. In this case, a belief in an engaged God may promote a more generalized distrust of science that could reduce the motivation to be vaccinated while turning over control of one’s life to the hands of a loving deity, an apparent form of divine fatalism.

The role of religious leaders during the COVID-19 pandemic and previous health crises has also been noted across scientific fields and it reveals complicated patterns [21]. Religious leaders often reach marginalized individuals (i.e., migrants, racial/ethnic minorities, and elderly persons), many of whom are unable to readily access health information through conventional routes due to language barriers or community distrust in public health. Consequently, one upside of religious leadership is found in the ability of pastors to serve as a conduit between health experts and their congregants, thereby dispelling misinformation such as conspiracy theories that COVID-19 is a hoax or that vaccines are a stealth weapon [21]. Yet, at the same time, some religious leaders have been known to persuade their congregants to engage in risky health-related behaviors using narratives of unshakable faith, skepticism toward science, and the endorsement of specious claims, including those promoted by politicians with whom they sympathize [20,22,23,24].

Individual autonomy during the pandemic has been a vocal concern expressed by political constituencies and religious groups alike in the United States, particularly in conservative circles. In relation to the coronavirus, religious freedom and scientific skepticism have been at the forefront of vaccine resistance and precautionary measure discourse (i.e., masking, social distancing, and lockdown measure adherence) [20,24,25,26]. In recent years, the U.S. has seen a rise in Christian nationalism through the political transformation of some conservative Protestants, namely, those who supported Donald Trump. The former U.S. president, in turn, advanced conservative Protestant religious ideology through the selection of conservative Supreme Court justices and utilized legal avenues in an attempt to overturn congregational closures during lockdown periods [26]. Additionally, both pre-pandemic and post-pandemic studies have shown that conservative Protestants or those who identify with the Republican Party are more skeptical of scientific facts [27,28]. In recent years, the authority of scientific leaders such as Anthony Fauci has been disparaged through public narratives that express skepticism towards science and those who practice it [28]. It is worth noting that, contrary to some media narratives, few racial or ethnic differences in skepticism toward COVID-19 exist, and the differences that do remain indicate less skepticism among Black and Asian individuals relative to Hispanic and white individuals, especially white Christian nationalists [20,29,30]. Further, in the spring of 2020 as COVID-19 cases first began to rise, most local officials across political parties and religious groups reported engaging in recommended personal health behaviors and supported the delayed reopening of closed institutions [31]. While our study cannot address this wide range of factors, Christian nationalism has been documented as a dominant influence among elected officials’ pandemic-related actions, particularly where resistance to public health guidance is concerned. Local officials with strongly religious nationalist views were vocal in prioritizing personal and religious freedom from government “intrusion” while also, in somewhat of a paradox, enthusiastically endorsing the robust presence of religion in government [31,32,33].

With this empirical background in mind, our investigation is guided by conceptual insights about the broadly anti-intellectual and anti-scientific tendencies that are evident within large swaths of conservative Protestantism. Previous research has revealed that deficits in verbal ability and scientific literacy exhibited within conservative Protestantism are generated and sustained through (1) an anti-intellectual ideology that privileges scripture over scientific knowledge and (2) social insularity through within-group bonding and less bridging with “outsiders” [12,13]. These factors can create a subcultural echo chamber that protects and even magnifies these tendencies. A community that is inclined to exhibit scientific skepticism would be expected to hesitate or refuse to engage in scientifically valid disease containment measures. These tendencies are not without potentially adverse consequences, as such an orientation can quickly become a vector for a robust and sustained contagion that is harmful not only to the subculture itself but to the broader society within which it is situated. Does this concern align with conservative Protestant pandemic perceptions and practices? To address this question, we turn to empirical data collected from a nationally representative sample of U.S. residents during the pandemic.

## 2. Materials and Methods

### 2.1. Data

Our study of denominational differences in risky pandemic lifestyles utilizes data from the 2021 Crime, Health, and Politics Survey (CHAPS). The primary purpose of CHAPS is to document the social causes and consequences of health and well-being in the United States during the coronavirus (COVID-19) pandemic. CHAPS is based on a national probability sample of 1771 community-dwelling adults aged 18 and over living in the United States. Respondents were sampled from the National Opinion Research Center’s (NORC) AmeriSpeak© panel, which is representative of households from all 50 states and the District of Columbia (https://amerispeak.norc.org/Documents/Research/AmeriSpeak%20Technical%20Overview%202019%2002%2018.pdf accessed on 25 November 2022). Sampled respondents were invited to complete the online survey in English between 10 May 2021 and 1 June 2021. The data collection process yielded a survey completion rate of 30.7% and a weighted cumulative response rate of 4.4%. The weighted cumulative response rate, which considers all panel recruitment and retention rates, is the overall survey response rate that accounts for survey outcomes in all response stages, including the panel recruitment rate, panel retention rate, and survey completion rate. It is weighted to account for the sample design and differential inclusion probabilities of sample members. Our cumulative response rate is within the typical range (4–5%) of high-quality general population surveys (see https://www.pewresearch.org/politics/2021/05/17/scope-of-government-methodology/ accessed on 25 November 2022). The multistage probability sample resulted in a margin of error of ±3.23% and an average design effect of 1.92. Margin of error is defined as half the width of the 95% confidence interval for a proportion estimate of 50% adjusted for design effect. A figure of ±3.23% is therefore the largest margin of error possible for all estimated percentages based on the study sample. A margin of error of ±3.23% at the 95% confidence level means that if we fielded the same survey 100 times, we would expect the result to be within 3.23% of the true population value 95 times. A margin of error of 3.00 is considered very good [34]. The average design effect is the variance under the complex design divided by the variance under a simple random sampling design of the same sample size. The design effect is variable-specific, and the reported value is the average design effect calculated for a set of key survey variables. Design effects account for deviations from simple random sampling with a 100% response rate. A design effect of 1.92 is very good because it means that the variance is only about twice as large as would be expected with simple random sampling [35]. The median self-administered web-based survey lasted approximately 25 min. All respondents were offered the cash equivalent of $8.00 for completing the survey, which is on the more lucrative end of the incentive spectrum for a survey of this duration. The survey was reviewed and approved by the institutional review boards at NORC and the lead author’s university. Informed consent was obtained from all participants.

### 2.2. Measures

#### 2.2.1. Dependent Variable I: Pandemic Threat Perceptions

We measure perceived pandemic threat as the mean response to four items. Respondents were asked to indicate the extent to which they agree or disagree with the following statements about the coronavirus (COVID-19) pandemic: (a) “The coronavirus pandemic is a major threat to public health in the United States.” (b) “The coronavirus pandemic is a major threat to your personal health.” (c) “The coronavirus pandemic is a major threat to the economy in the United States.” (d) “The coronavirus pandemic is a major threat to your personal financial situation.” Responses to these items ranged from (1) strongly disagree to (5) strongly agree. All items were coded so that higher values would indicate greater pandemic threat. An exploratory principal components analysis with varimax rotation produced a single component for the four items (eigenvalue = 2.37), with loadings ranging from 0.67 to 0.86. A reliability analysis also suggested good internal consistency for four items (α = 0.77).

#### 2.2.2. Dependent Variable II: Healthy Pandemic Lifestyles

We measure healthy pandemic lifestyles as the mean response to four items. Respondents were asked to indicate how often during the coronavirus (COVID-19) pandemic they (a) “attended indoor gatherings with more than 10 people”, (b) “used hand sanitizer to kill germs after being in public places”, and (c) wore “a mask or other face covering in public places.” Responses to these items ranged from (1) never to (5) always, with reverse-coding for the indoor gatherings indicator. Respondents were also asked (d) if they had been “vaccinated for the coronavirus (COVID-19).” Responses to this item were coded (1) yes and (0) no. To account for metric differences, each of these items was standardized before indexing. All items were coded so that higher values would indicate healthier pandemic behavior. An exploratory principal components analysis with varimax rotation produced a single component for the four items (eigenvalue = 1.97), with loadings ranging from 0.65 to 0.83. A reliability analysis also suggested adequate internal consistency for four items (α = 0.65).

#### 2.2.3. Key Predictor Variable: Religious Affiliation

We measure religious affiliation with six dummy variables. These variables capture (a) conservative Protestants (those who reported being Protestant and evangelical/born again), (b) moderate Protestants (those who reported being Protestant without being evangelical/born again), (c) Catholics, (d) other Christians (e.g., those who reported being Mormon, Orthodox, or “just Christian”), (e) other religions (e.g., Jews, Buddhists, and Muslims), and (f) non-affiliates (those with no religious affiliation, including atheists and agnostics).

#### 2.2.4. Control Variables

Background variables include *age* (continuous years), *gender* (1 = female; 0 = male), *race*/*ethnicity* (dummy variables for non-Hispanic black, Latino, and other race or ethnicity, with non-Hispanic white serving as the reference), *nativity status* (1 = US-born; 0 = otherwise), *southern residence* (1 = South; 0 = otherwise), *rural residence* (1 = rural; 0 = otherwise); *college degree* (1 = four-year college degree or higher; 0 = otherwise), *employment* (1 = employed; 0 = otherwise), *annual household income* (1 = <$10,000 to 9 = ≥$150,000), *marital status* (1 = married; 0 = otherwise), children (1 = presence of a child under age 18; 0 = otherwise, and *Republican Party* (1 = Republican; 0 = otherwise). Finally, *financial strain* is measured as the mean response to three items. Respondents were asked to indicate the extent to which their household had trouble paying (a) for “needed health care”, (b) “monthly bills”, and (c) for “food.” Response categories for these items ranged from (1) never to (5) all the time. An exploratory principal components analysis with varimax rotation produced a single component for the three items (eigenvalue = 2.39), with loadings ranging from 0.83 to 0.93. A reliability analysis also suggested excellent internal consistency for three items (α = 0.89).

### 2.3. Analysis

Due to listwise deletion of missing data, our analytic sample size was reduced from 1771 to 1746. Post-stratification weights were used in subsequent analyses to reduce sampling error and non-response bias. NORC developed post-stratification weights for CHAPS via iterative proportional fitting or raking to general population parameters derived from the *Current Population Survey* (https://www.census.gov/programs-surveys/cps/data.html accessed on 25 November 2022). These parameters included age, gender, race/ethnicity, education, and several interactions (age*gender, age*race, and gender*race).

Our analyses begin with weighted descriptive statistics for all study variables, including variable ranges, sample means, and standard deviations (Table 1). In Table 2, we model pandemic threat perceptions and healthy pandemic behaviors as a function of religious affiliation and background variables (Models 1 and 4). These models assess baseline religious affiliation differences in perceived pandemic threat and pandemic lifestyles. Models 2 and 5 of Table 2 add interaction terms to formally test whether the religious affiliation differences observed in Models 1 and 4 vary by southern residence. Models 3 and 6 add political party identification to Models 1 and 4 to account for the fact that conservative Protestants are often Republicans. These models assess the degree to which religious affiliation differences are observed net of political leanings. Finally, in Table 3, we model the individual pandemic threat and pandemic lifestyle items to assess whether particular items might be driving any observed religious affiliation differences in our focal indices.

## 3. Results

### 3.1. Descriptive Analyses

According to Table 1, which features descriptive statistics based on an analyses of the weighted data, the average respondent perceived the pandemic to be a “major threat” and reported engaging in healthy pandemic lifestyles (that is, behaviors that comply with public health recommendations). Where religious affiliation is concerned, the sample included conservative Protestants (22%), moderate Protestants (12%), Catholics (20%), other Christians (16%), respondents of other religious faiths (5%), and respondents with no religious affiliation (25%). The average age of the sample was 48 years. The race and ethnic composition of the sample included non-Hispanic whites (63%), non-Hispanic blacks (11%), Latinos (17%), and respondents of other races and ethnicities (9%). (It is worth noting that these racial/ethnic proportions are affected somewhat by the weighting techniques.) Few respondents reported being born in another country (10%) or living in a rural area (17%). However, an appreciable proportion of respondents reported living in the South (38%), and just over one third indicated having a four-year college degree or higher (35%). Nearly six in ten of our respondents reported being employed full-time or part-time (59%). The average respondent also reported an annual household income between $50,000 and $74,999 and “rarely” having any difficulty paying for health care, food, or other bills. Where family characteristics are concerned, slightly over half of our respondents reported being married (52%), and a small proportion reported having a child under the age of 18 present in their household (17%). Finally, over one third of respondents reported being a member of the Republican Party (38%).

### 3.2. Regression Analyses

In Model 1 of Table 2, we find that while moderate Protestants, Catholics, respondents of other religious faiths, and respondents with no religious affiliation tend to report higher levels of pandemic threat than conservative Protestants, other Christians and conservative Protestants tend to report similar levels of threat perception. In Model 4 of Table 2, we observe a slightly more uniform profile, with moderate Protestants, Catholics, other Christians, respondents of other religious faiths, and respondents with no religious affiliation exhibiting healthier pandemic lifestyles than conservative Protestants. We note that these patterns are persistent with adjustments for age, gender, race/ethnicity, nativity status, southern residence, rural residence, education, employment status, household income, financial strain, marital status, and the presence of children.

In Models 2 and 5 of Table 2, we formally tested whether the religious affiliation differences observed in Models 1 and 4 varied by southern residence. In both pandemic threat and pandemic lifestyles regression models, none of the interaction terms reached statistical significance. These findings suggest that the pandemic attitudes and behaviors of conservative Protestants are comparable in southern states and in states located in other regions of the country.

In Models 3 and 6 of Table 2, we added political party identification to Models 1 and 4. In Model 3, the levels of pandemic threat reported by moderate Protestants and respondents with no religious affiliation are no longer different from conservative Protestants. Although we observed considerable attenuation in the original coefficients from Model 1 to Model 3, Catholics and respondents of other faiths continue to report higher levels of perceived pandemic threat than conservative Protestants. In Model 6, the pandemic lifestyles reported by other Christians are no longer different from conservative Protestants. Again, with notable attenuation, moderate Protestants, Catholics, respondents of other religious faiths, and respondents with no religious affiliation continue to exhibit healthier pandemic lifestyles than conservative Protestants. Considered together, these patterns suggest that one reason why conservative Protestants tend to be less concerned about the pandemic and less likely to comply with public health recommendations is that they also tend to identify with the Republican Party. Models 3 and 6 clearly show that Republicans tend to report lower levels of pandemic threat and riskier pandemic lifestyles than respondents who do not identify as Republicans. Although Republican identification is associated with conservative Protestantism and with pandemic attitudes and behaviors, we note that two thirds of the original religious affiliation differences observed in Models 1 and 4 persisted with adjustments for political party in Models 3 and 6.

In Table 3, we model the individual pandemic threat and pandemic lifestyle items. The results for the pandemic threat items reveal that original religious affiliation differences observed in Model 1 of Table 2 were entirely driven by perceived pandemic threats to public health and personal health. In fact, we failed to observe any religious affiliation differences in the perceived pandemic threats to the economy or to personal finances. The results for the pandemic lifestyle items show that the original religious affiliation differences observed in Model 4 of Table 2 were mostly driven by religious differences in attending indoor gatherings with more than 10 people, using masks, and receiving vaccinations. The original difference between other Christians and conservative Protestants was mostly attributable to differences in attending indoor gatherings. While other Christians attended indoor gatherings less frequently than conservative Protestants, these groups were statistically indistinguishable in the cases of masks and vaccinations. There were no religious affiliation differences in the use of hand sanitizer.

### 3.3. Supplemental Analyses

In supplemental analyses (not shown, available by request), we tested whether the religious affiliation differences in pandemic attitudes and behaviors also varied by race/ethnicity, education, and rural residence. In this analysis, we tested 30 total interactions, and only 4 reached statistical significance in the case of pandemic threat. The difference between other Christians and conservative Protestants was more pronounced among respondents living in rural areas. In fact, the levels of perceived pandemic threat were similar for other Christians and conservative Protestants among respondents living in non-rural areas. The opposite pattern was observed for the difference between respondents with no religious affiliation and conservative Protestants. In other words, respondents with no religious affiliation reported higher levels of pandemic threat than conservative Protestants in non-rural areas, but not in rural areas. With respect to educational variations, differences in perceived pandemic threat between conservative Protestants and respondents of other religious faiths were more pronounced among respondents with college degrees than without. Similarly, respondents with no religious affiliation reported higher levels of pandemic threat than conservative Protestants among respondents with college degrees, but not among lesser-educated respondents. In the case of perceived pandemic threat, differences involving moderate Protestants and Catholics were invariant across these particular subgroups, and no variations by race/ethnicity were observed for any of the religious affiliation differences. We also failed to observe any subgroup variables in any religious affiliation differences in pandemic lifestyles.

## 4. Discussion

This study set out to examine religious variations in pandemic threat perceptions and healthy pandemic lifestyles with a focus on the suspected distinctions exhibited by conservative Protestants in the U.S. We hypothesized that, on average, conservative Protestants would exhibit lower levels of perceived pandemic threat and riskier pandemic lifestyle behaviors compared with other religious groups and their non-religious peers. So, during the COVID-19 pandemic, were conservative Protestants less concerned about the threats posed by this event than other religious groups and the nonreligious? In a word, yes. On average, moderate Protestants, Catholics, other religious adherents, and those reporting no religious affiliation reported more robust pandemic threat perceptions than conservative Protestants. These differences withstood controls for a wide range of confounding factors. It is important to note that it was difficult to categorize other Christians (some of whom are “just Christian” nondenominational religious conservatives, along with Mormons and Orthodox Christians), who were generally on par with conservative Protestants in terms of pandemic threat perceptions. Were conservative Protestants more likely to engage in riskier pandemic lifestyles? Again, yes. Moderate Protestants, Catholics, other Christians, those affiliated with other religious faiths, and respondents with no religious affiliation exhibited healthier pandemic lifestyles than conservative Protestants. Here again, these observed differences generally persisted with adjustments for key confounders, including age, gender, race/ethnicity, nativity status, southern residence, rural residence, education, employment status, household income, financial strain, marital status, and the presence of children. Our inclusion of possible confounding variables in statistical models reduces the threat of spuriousness and improves confidence in the soundness of our results. In short, conservative Protestant separation from mainstream culture could contribute to a distinct health liability during the pandemic. To facilitate an understanding of our key results, we summarize and discuss their larger implications in what follows.

Distinctive attitudes and practices exhibited by conservative Protestants did not vary appreciably in southern versus non-southern states. This point is vitally important. We began this paper quoting the words of a southern Republican governor who made headlines in 2021 by asserting that a focus on eternal life and otherworldly considerations diminishes fear about this-worldly threats, including a pandemic among Christian believers. We wondered if this sentiment was a distinctively southern expression of Christian conservatism. It is not. We observed quite broadly that conservative Protestants were less inclined to perceive the pandemic as threatening or to engage in healthy pandemic behavior. This finding is noteworthy because there have been discussions for some time about internal variability among conservative Protestants, with some charging and others contesting that such internecine fissures break down along regional lines [36]. Where two key forms of COVID-19 response are concerned (pandemic threat perceptions and healthy pandemic lifestyles), we find no evidence of regional variation. Republican Party identification, however, tells another story. We find some attenuation of the effects for conservative Protestant affiliation when controlling for party identification. Roughly one third of the effects of conservative Protestantism for each of our two sets of pandemic outcomes are a product of Republican Party identification, which tends to be the party of choice among conservative Protestants. Therefore, the independent effects of conservative Protestant affiliation remain robust, but are intertwined with this key political factor.

When parsing out specific outcomes in our pandemic threat and healthy pandemic lifestyles indexes, some interesting findings emerged. Religious differences in perceived pandemic threat were a product of threat perceptions related to public health and personal health. Perceptions about widespread economic threats or vulnerability related to personal finances due to the pandemic were not associated with religious affiliation. This finding suggests that *health-related* appraisals of pandemic threat were the wedge that distinguished conservative Protestants from others. This finding adds to previous research that reveals health resources to be less robust in conservative Protestant communities and other evidence of anti-science orientations among conservative Protestants [3,37]. It is possible that survey respondents, regardless of their denominational affiliation, understood the economic and personal finance fallout of the pandemic as an inarguable effect of this event because it was never publicly challenged by religious or political elites. The key point of this denominational distinction is the perceived threat to human health and willingness to follow the guidance of infectious disease experts. On the latter point, conservative Protestants distinguish themselves as lacking faith in medical science. While this “lacking faith” characterization may seem like a pun, it is intended to underscore the distinctively low levels of trust that conservative Protestants exhibit in medical science. This finding should not be altogether surprising, as it aligns closely with conservative Protestants’ skeptical view of science in general [37] and their opposition to policy recommendations advanced by scientists even when empirical findings are not scientifically disputed [38]. Whether this broadly anti-science mentality or opposition to science-based policy recommendations undermined compliance with public health recommendations is a question we cannot adjudicate with our data. The specific survey measures that principally yielded religious differences concerning pandemic lifestyles were attendance at indoor gatherings of more than 10 persons (possibly worship service attendance), wearing masks, and getting vaccinated. Here again, there were some convergences between conservative Protestants and other Christians (likely, nondenominational adherents and other conservative Christians such as Mormons) concerning mask use and vaccination uptake. However, this skeptical posture toward science in the name of religious orthodoxy likely contributed to behaviors that medical research says can prolong a pandemic.

Several implications stem from these findings. First, while conservative Protestants seem to have a pronounced and widespread skepticism toward pandemic and public health recommendations, there is considerable diversity within this religious subculture on some hot-button social issues. Gender ideologies among conservative Protestant married couples and child-rearing practices among parents within this faith tradition are diverse and complex [39,40]. More to the point, there is evidence of better health outcomes among evangelicals, one of the more world-engaging subgroups among conservative Protestants, as compared with fundamentalists and Pentecostals [3]. Additionally, some leading evangelicals directly challenged religious conservative misinformation efforts undertaken by prominent conservative Protestants during the pandemic [30]. Therefore, it is possible that public health information might be better received in evangelical (world-engaging) congregations than in fundamentalist (world-condemning) and Pentecostal (faith-healing) congregations. Additionally, some high-profile evangelicals are clearly receptive to the lessons learned from pandemic science. So, perhaps the building of selective alliances and strategic dialogues with the evangelical wing of conservative Protestantism could provide a path forward for public health advocacy.

Second, public health promotion should likely occur across institutional venues, such as religion and politics. While it is difficult to imagine a future world in which masks and vaccinations would not be a culturally charged issue given current polarization, efforts could be made to describe masks and vaccines as forms of protection for the “bodily temples” of devout conservative Christians (conservative religious messaging) and as an “insurance policy” designed to foster a quicker economic recovery for the nation at large (conservative political messaging). Given the intertwining of religious orthodoxy and conservative politics in relation to the pandemic, holistic messages that affirm values held by these constituencies could go a long way toward minimizing resistance to empirically substantiated practices, such as masking and vaccination uptake, particularly during times of high disease transmission. One optimistic note is that hand sanitizer use did not exhibit significant religious variations. The wearing of masks became a political and then religious lightning rod, but hand sanitizer use seemed to be interpreted quite simply as good hygiene. Perhaps there is a future in which masks could be viewed similarly as a hygienic asset during periods of high infection or circumstances of elevated risk.

Finally, theories of conservative Protestantism as a religious subculture have warned against the “angel–demon” dichotomy that alternatively valorizes or vilifies members of this faith tradition [41,42]. Conservative Protestant opposition to some facets of mainstream culture has tended to provoke one of these two reactions. However, where health outcomes are concerned [3], much like those regarding racial inequality, conservative Protestants do not have an admirable track record [43,44,45,46]. So, while stopping short of demonizing conservative Protestantism, which is not our role as social scientists, a valid criticism of significant health vulnerabilities grounded in empirical science (even if that science is questioned by the religious practitioners themselves) is certainly warranted, especially when the health and welfare of people within and outside the subculture is at stake. Criticism with an openness to positive change need not involve vilification.

Our study is marked by various limitations, all of which can be addressed through additional research. First, as a cross-sectional investigation, we cannot directly examine causal mechanisms and must presume causal order (religious affiliation predictors and pandemic response outcomes). It is theoretically possible that religious groups will be realigned and even reconstitute themselves in the wake of large historical upheavals such as a pandemic. So, a pandemic could have effects on religious composition rather than religious affiliation being a static, unidirectional driver of health perceptions and behaviors. Religious adherents who disagree with the dominant conservative Protestant skepticism about the pandemic might find themselves driven out of congregations that previously welcomed them or may elect to exit on their own accord. Longitudinal research is needed to determine how the composition of religious groups might be affected by the pandemic and the swirl of political ideologies surrounding it. The public health consequences of such changes could be substantial, as opportunities for health-related outreach and education could become increasingly rare among religious groups that become more ideologically stringent and even stridently anti-science over time. We have presumed that religion is a contributing factor that shapes pandemic attitudes and behaviors, which is warranted for our investigation. However, the prospect for religious sorting or realignment as a pandemic consequence should not be ignored.

Second, we need to learn more about interpretive processes that undergird pandemic skepticism among conservative Protestants. Qualitative research (in-depth interviews and focus groups with adherents, ethnographic fieldwork in congregations, and textual analyses of conventional and social media messaging) is needed to examine processes of legitimation (ideological, theological, etc.) in conservative Protestant faith communities. Our study cannot address this important issue. Conservative Protestant interpretive schemas in relation to the pandemic remain an important black box that, if not well understood by researchers, could complicate public health promotion efforts. Additionally, there is evidence of scripturally based conservative Protestant messaging in favor of pandemic health precautions at the elite level that could be reflected in diversity at the grassroots [30].

Finally, some of our supplementary analyses highlight the importance of considering exogenous factors that intersect with denominational distinctions in pandemic response. We did not observe spatial variations at the regional level. South versus non-South distinctions were absent, so the effects of conservative Protestant affiliation were not unique to the U.S. South. However, more localized spatial differences seem to influence pandemic perceptions and behaviors for conservative Protestants. Therefore, split-sample analyses of conservative Protestant effects in rural versus urban locales seem promising.

Additional lines of inquiry could be fruitfully pursued, including ecological (county-level) COVID-19 mortality variations in relation to conservative Protestant concentration, which would extend early pandemic research on generic religious differences in health directive compliance [19]. It could also be beneficial to compare conservative Protestant support of non-pandemic health behaviors (e.g., shingles vaccination, birth control, plastic surgery) with pandemic health perspectives (e.g., vaccine response, COVID-19 treatment). Future research would also gain much from considering how persons of faith, religious communities, and interfaith organizations negotiate religious convictions and scientific facts with respect to the COVID-19 pandemic. Perhaps faith and science are not locked in a zero-sum contest of oppositional perspectives. There may be nuances to uncover among conservative religious groups, especially those whose education has given them greater scientific literacy. Another intriguing prospect for future research would entail examining Christian nationalist or prosperity gospel views as possible mediators of conservative Protestant pandemic threat perceptions and pandemic lifestyle behaviors, given the rise of these ideologies within conservative Protestantism [32,33]. Prosperity gospel theology, in particular, emphasizes health and wealth as divine blessings for obedience to God’s commandments. Because religious and scriptural teachings generally do not promote vaccine and medical resistance in many faith traditions (e.g., Judaism, Christianity, Islam), the rise of Christian nationalism during the pandemic could have paved the way for conservative Protestants to lean more on individualistic faith and political narratives that contributed to selective health science skepticism [8,23]. Additionally, the possible effects of religious attendance should be scrutinized. More frequent attendance at worship services, commonly observed in conservative faith traditions, could indicate more robust religious network embeddedness and greater exposure to pandemic messaging from congregational leaders and peers. In the end, we would do well to recognize that religious expression and involvement have a complex connection to the pandemic, one certainly meriting continued study using a variety of methodologies, data sources, and conceptual approaches.

## 5. Conclusions

This study has provided compelling cross-sectional evidence of the association between conservative Protestant affiliation and skepticism toward the COVID-19 pandemic. That skepticism was expressed in two forms: pandemic threat perceptions and pandemic lifestyles. For both outcomes, conservative Protestants distinguished themselves from most or, in some cases, all other religious groups and their non-religious counterparts. Specifically, conservative Protestant respondents were less likely to perceive the pandemic as a threat to human and social welfare, principally due to their skepticism about the community and personal health threats posed by the coronavirus. In addition, conservative Protestants were less likely to engage in healthy pandemic mitigation measures such as social gathering restriction adherence, masking to prevent the spread of infection, and vaccination use. Our study contributes to a growing body of research that has underscored the health vulnerabilities associated with some forms of religious involvement, in this case, conservative Protestant affiliation. Additional work needs to be conducted to dissect the specific sources of pandemic skepticism, including a possible prioritization of scriptural resources and otherworldly sensibilities over scientific reason and this-worldly considerations. Bridging the gaps between faith and reason as competing perspectives in our current times is among the most pressing challenges faced by those concerned about public health. Although it is difficult to believe in our polarized moment, faith and reason may be redefined as complementary facilitators of good health rather than competing worldviews.

## Figures and Tables

**Table 1 healthcare-11-00582-t001:** Weighted descriptive statistics (CHAPS 2021).

	Range	Mean	Standard Deviation
Pandemic Threat Perceptions	1 to 5	3.65	0.86
Healthy Pandemic Behaviors	−2.67 to 0.84	−0.01	0.71
Conservative Protestant	0 to 1	0.22	
Moderate Protestant	0 to 1	0.12	
Catholic	0 to 1	0.20	
Other Christian	0 to 1	0.16	
Other Religion	0 to 1	0.05	
No Affiliation	0 to 1	0.25	
Age	18 to 94	48.03	17.52
Female	0 to 1	0.52	
Non-Hispanic White	0 to 1	0.63	
Non-Hispanic Black	0 to 1	0.11	
Latino	0 to 1	0.17	
Other Race/Ethnicity	0 to 1	0.09	
US-Born	0 to 1	0.90	
Southern Residence	0 to 1	0.38	
Rural Residence	0 to 1	0.17	
College Degree	0 to 1	0.35	
Employed	0 to 1	0.59	
Household Income	1 to 9	5.51	2.29
Financial Strain	1 to 5	1.70	0.93
Married	0 to 1	0.52	
Presence of Children	0 to 1	0.17	
Republican	0 to 1	0.38	

**Table 2 healthcare-11-00582-t002:** Weighted regressions of pandemic threat perceptions and healthy pandemic behaviors (CHAPS 2021).

	1. Pandemic Threat Perceptions		2. Pandemic Threat Perceptions		3. Pandemic Threat Perceptions		4. Healthy Pandemic Behaviors		5. Healthy Pandemic Behaviors		6. Healthy Pandemic Behaviors	
Moderate Protestant	0.17	*	0.16		0.08		0.28	***	0.28	**	0.20	**
Catholic	0.29	**	0.19		0.19	*	0.25	***	0.19	*	0.17	*
Other Christian	0.05		0.04		−0.06		0.18	*	0.12		0.09	
Other Religion	0.41	**	0.25		0.25	*	0.46	***	0.38	**	0.32	**
No Affiliation	0.16	*	0.18		−0.02		0.31	***	0.33	***	0.14	*
Mod. Prot.* South			0.01						0.01			
Catholic* South			0.28						0.18			
Other Christ.* South			0.04						0.15			
Other Relig.* South			0.42						0.21			
No Aff.* South			−0.12						−0.10			
Age	0.01	***	0.01	**	0.01	***	0.01	***	0.01	***	0.01	***
Female	0.07		0.07		0.03		0.15	**	0.14	**	0.11	*
Non-Hispanic Black	0.36	***	0.37	***	0.13		0.27	***	0.26	***	0.06	
Latino	0.10		0.09		0.02		0.03		0.03		−0.04	
Other Race/Ethnicity	0.16		0.15		0.12		0.11		0.10		0.07	
US-Born	−0.22	*	−0.24	*	−0.19		−0.09		−0.09		−0.06	
Southern Residence	0.09		0.04		0.10		−0.06		−0.10		−0.05	
Rural Residence	0.11		0.11		0.12		−0.03		−0.04		−0.02	
College Degree	0.12	*	0.11	*	0.09		0.14	**	0.14	**	0.11	*
Employed	−0.02		−0.02		−0.01		−0.04		−0.03		−0.03	
Household Income	−0.03		−0.03		−0.01		0.02		0.02		0.01	
Financial Strain	0.13	***	0.13	**	0.12	**	−0.05	*	−0.05	*	−0.06	*
Married	−0.05		−0.05		0.01		−0.02		−0.02		0.03	
Presence of Children	0.09		0.08		0.11		−0.05		−0.05		−0.02	
Republican					−0.51	***					−0.46	***

(* *p* < 0.05, ** *p* < 0.01, and *** *p* < 0.001).

**Table 3 healthcare-11-00582-t003:** Weighted regressions of pandemic threat perceptions and healthy pandemic behaviors (CHAPS 2021).

	Pandemic Threat Perceptions								Healthy Pandemic Behaviors							
	Threat to Public Health		Threat to Personal Health		Threat to Economy		Threat to Personal Finances		Public Gatherings		Hand Sanitizer		Masks/Face Coverings		Coronavirus Vaccination	
Moderate Protestant	0.33	**	0.25	*	0.08		0.02		0.45	***	0.17		0.29	**	0.52	*
Catholic	0.40	**	0.44	***	0.09		0.21		0.35	**	0.11		0.28	**	0.78	**
Other Christian	0.21		0.15		−0.15		−0.03		0.36	**	0.03		0.17		0.44	
Other Religion	0.67	***	0.66	***	0.06		0.23		0.60	***	0.26		0.48	***	1.29	**
No Affiliation	0.44	***	0.28	*	−0.02		−0.08		0.69	***	−0.05		0.32	**	0.59	**

(* *p* < 0.05, ** *p* < 0.01, and *** *p* < 0.001). All models adjust for age, gender, race/ethnicity, nativity status, southern residence, rural residence, college degree, employment status, household income, financial strain, marital status, and presence of children.

## Data Availability

Data available on request due to restrictions (e.g., privacy or ethical). The data are not publicly available due to data held in a private repository.

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
