# Peer review of "Fear God, Not COVID-19: Is Conservative Protestantism Associated with Risky Pandemic Lifestyles?"

_healthcare, 2023, doi:10.3390/healthcare11040582_

Round 1
Reviewer 1 Report
“Fear God, Not COVID” is a very good article; the authors offered a good discussion of relevant literature, conducted and presented a significant empirical study or how religious outlooks affect the ways the pandemic is viewed, and provided a solid discussion of their research findings. Overall, the article documents and discusses insightfully the link between pandemic skepticism and pandemic risk-taking among conservative Protestants. I recommend it for publication.
My primary concern is while the authors’ findings suggest that people should challenge the tendency to oppose faith and science, much of their analysis reinforces this unhealthy dichotomy. Note that the title “Fear God, Not COVID” can be read as suggesting that we consider the question: Should we fear God, or should we fear COVID-19? The overall focus of the article is that we should fear COVID-19 more than we fear God.
The article does not report on the literature that documents how many people turned to faith during the COVID-19 pandemic and found personal comfort, support in their religious community, and motivation to care for those afflicted with COVID-19 disease and to commit to seeking ways, based on the best scientific findings, to stop the spread of the disease. The article highlights Christian theological stances that are based on fear of God and what the authors identity as “otherworldly and individualistic convictions” (74). It does not report on people of faith and faith communities who responded to the pandemic based on a commitment to respect and care for all persons as persons and a commitment to work for the common of society. Perhaps in their Discussion of their findings the authors could note this limitation of their study and call for studies that explore how persons of faith, faith communities and interfaith organizations in many instances opted for a both-and (drawing insight from both faith and science) rather than an either/or (faith or science) stance in responding to COVID-19.
In their discussion of the implications of the study the authors call for “public health promotion.” I think they could also call for religious education in faith communities that explores fruitful ways of relating faith and science.
The authors call for future research to examine “Christian nationalist and prosperity gospel views” as possible mediators of conservative Protestant pandemic threat perception. A careful review of events over the past two years would show that this suggestion is questionable. COVID-19 is a global threat requiring a global response. Christian nationalists have tended to block rather than foster fruitful reflection and action in response to the pandemic. Perhaps one of the clearest examples of this is Christian nationalists in the United States referring to the pandemic as the “China virus” and blaming a foreign country for the pandemic, and then encouraging US Christians to turn inward, block out the rest of the world through walls and travel and trade restrictions, and care those in need here in the United States and let the rest of the world deal with the pandemic as best they can on their own. Somewhat similarly, the prosperity gospel affirms rather than challenges the hyper-individualism found in conservative Protestantism and has been used in some instances to foster a kind of survival-of-the-fittest version/distortion of Christian faith.
Overall, the social scientific analysis in the article is excellent, but the article’s theological analysis lacks depth and the authors fail to provide a full and nuanced account of how Christian responses to the COVID-19 pandemic have in some cases helped while in other instances they hindered people’s and community’s abilities to respond to the pandemic. Might the authors revise slightly their Introduction and Discussion sections in order to offer a more nuanced theological analysis of Christian responses to the pandemic?
Author Response
REVIEWER 1
“Fear God, Not COVID” is a very good article; the authors offered a good discussion of relevant literature, conducted and presented a significant empirical study or how religious outlooks affect the ways the pandemic is viewed, and provided a solid discussion of their research findings. Overall, the article documents and discusses insightfully the link between pandemic skepticism and pandemic risk-taking among conservative Protestants. I recommend it for publication.
AUTHOR RESPONSE: We appreciate your thorough comments and recommendation for this article to be published.
My primary concern is while the authors’ findings suggest that people should challenge the tendency to oppose faith and science, much of their analysis reinforces this unhealthy dichotomy. Note that the title “Fear God, Not COVID” can be read as suggesting that we consider the question: Should we fear God, or should we fear COVID-19? The overall focus of the article is that we should fear COVID-19 more than we fear God.
AUTHOR RESPONSE: The introduction and discussion sections have been revised to more accurately reflect the current literature and this study’s findings. Rather than reinforcing the opposition between faith and science, we now include perspectives related to a beneficial relationship between faith and science through the COVID-19 pandemic (e.g., congregational leaders’ endeavors to promote vaccinations). We do maintain that conservative Protestants have widely opposed COVID-19 public health and safety procedures, especially as the pandemic evolved, unlike many of their peers belonging to different faith traditions in the United States.
The article does not report on the literature that documents how many people turned to faith during the COVID-19 pandemic and found personal comfort, support in their religious community, and motivation to care for those afflicted with COVID-19 disease and to commit to seeking ways, based on the best scientific findings, to stop the spread of the disease. The article highlights Christian theological stances that are based on fear of God and what the authors identity as “otherworldly and individualistic convictions” (74). It does not report on people of faith and faith communities who responded to the pandemic based on a commitment to respect and care for all persons as persons and a commitment to work for the common of society. Perhaps in their Discussion of their findings the authors could note this limitation of their study and call for studies that explore how persons of faith, faith communities and interfaith organizations in many instances opted for a both-and (drawing insight from both faith and science) rather than an either/or (faith or science) stance in responding to COVID-19.
AUTHOR RESPONSE: The introduction section now includes references to findings of religious support in the pandemic crisis, namely, how some religious congregations (many of which were not conservative Protestant) encouraged public health practices in the early stages of the pandemic. Our introduction now includes mentions of data retrieved in the first year of the pandemic (2020) that highlight the support of many religious individuals for coronavirus health and safety measures. Additionally, our discussion now also includes a call for future research to explore the relationship between persons of faith, faith communities, and interfaith organizations in negotiating religious beliefs and scientific facts with respect to COVID-19 pandemic health behaviors. We readily recognize that faith and science may not be viewed in a zero-sum fashion.
In their discussion of the implications of the study the authors call for “public health promotion.” I think they could also call for religious education in faith communities that explores fruitful ways of relating faith and science.
AUTHOR RESPONSE: The mention of religious education with a focus on scientific pathways is now included in the discussion section.
The authors call for future research to examine “Christian nationalist and prosperity gospel views” as possible mediators of conservative Protestant pandemic threat perception. A careful review of events over the past two years would show that this suggestion is questionable. COVID-19 is a global threat requiring a global response. Christian nationalists have tended to block rather than foster fruitful reflection and action in response to the pandemic. Perhaps one of the clearest examples of this is Christian nationalists in the United States referring to the pandemic as the “China virus” and blaming a foreign country for the pandemic, and then encouraging US Christians to turn inward, block out the rest of the world through walls and travel and trade restrictions, and care those in need here in the United States and let the rest of the world deal with the pandemic as best they can on their own. Somewhat similarly, the prosperity gospel affirms rather than challenges the hyper-individualism found in conservative Protestantism and has been used in some instances to foster a kind of survival-of-the-fittest version/distortion of Christian faith.
AUTHOR RESPONSE: The examination of Christian nationalist and prosperity gospel views as mediating factors on pandemic response threat perception would be well aligned with Christian nationalism’s and prosperity gospel’s promotion of individualism over collectivism. While conservative Protestant congregational responses to the pandemic differed over time (becoming increasingly critical as the pandemic wore on), the consensus that public health procedures (e.g., social distancing, vaccination, masking) were an infringement on religious rights was led by high-profile conservative—and often political—leaders, as mentioned in our introduction. Our introduction also includes a mention of media interference, which often publicized, disparagingly and encouragingly, conservative Protestant leaders and congregants who openly flouted health and safety procedures. The inclusion of references that detail congregational leaders who supported vaccine compliance has been interwoven throughout the introduction.
Overall, the social scientific analysis in the article is excellent, but the article’s theological analysis lacks depth and the authors fail to provide a full and nuanced account of how Christian responses to the COVID-19 pandemic have in some cases helped while in other instances they hindered people’s and community’s abilities to respond to the pandemic. Might the authors revise slightly their Introduction and Discussion sections in order to offer a more nuanced theological analysis of Christian responses to the pandemic?
AUTHOR RESPONSE: We have significantly revised the introduction and discussion sections of this article to reflect more nuanced theological standpoints with respect to the COVID-19 public health response. Our article’s primary focus is on conservative Protestants’ health behavior responses to the COVID-19 pandemic, not all Christian responses, which varied. Greater attention to these variations in public health support is now included. These variations are also included in our discussion section, which reiterates our results that moderate Protestants, Catholics, other Christians, respondents of other religious faiths, and respondents with no religious affiliation exhibited healthier pandemic lifestyles than conservative Protestants. A growing body of evidence, including our own data, highlights conservative Protestants’ lack of support for coronavirus pandemic safety measures regarding health behaviors. We do explicitly state that our role as social scientists is not to condemn conservative Protestants. But we hasten to acknowledge the potential consequences of the patterns we have observed.
Reviewer 2 Report
To start with, I would like to congratulate the authors of this very important matter. This study set out to examine religious variations in Coronavirus Pandemic threat perceptions and healthy pandemic lifestyles with a focus on suspected distinctions exhibited by conservative Protestants in the U.S. The authors approached the subject matter very well. Hence the work is both unique and original.
Authors looked at the scientific problem, but he needed to look at the problem individually (of course together with the method).
In my opinion, it would be worth preparing a traditional bibliography, because in its present form it consists of footnotes. My suggestions are exactly that, I leave it up to the authors to use them or not.
Looking at the work as a whole, it is well presented. The solution of the problem is well set out in the conclusion of the work.
It is very good that the authors call for additional research, including a possible prioritization of scriptural resources and otherworldly sensibilities over scientific reason and this-worldly considerations. Bridging the gaps between faith and reason as competing perspectives in our current times is among the most pressing challenges faced by those concerned about public health. Though it is difficult to believe in our polarized times, faith and reason may be redefined as complementary facilitators of good health rather than competing worldviews.
There are no direct studies on this subject. It raises no objections in terms of content and methodology. The statistical methods have been applied correctly. The use of charts and tables allows readers to follow the author's scientific proceedings. The pattern of proceedings presented in this way has led to conclusions that are consistent with the evidence and reasoning applied. In my opinion the paper is of highest value and therefore I would suggest publishing it.
Author Response
REVIEWER 2
To start with, I would like to congratulate the authors of this very important matter. This study set out to examine religious variations in Coronavirus Pandemic threat perceptions and healthy pandemic lifestyles with a focus on suspected distinctions exhibited by conservative Protestants in the U.S. The authors approached the subject matter very well. Hence the work is both unique and original.
AUTHOR RESPONSE: We appreciate your review, approval, and understanding of our study’s subject matter.
Authors looked at the scientific problem, but he needed to look at the problem individually (of course together with the method).
AUTHOR RESPONSE:
In my opinion, it would be worth preparing a traditional bibliography, because in its present form it consists of footnotes. My suggestions are exactly that, I leave it up to the authors to use them or not.
AUTHOR RESPONSE: We have organized a reference list in accordance with the American Sociology Association’s (as sociology is the authors’ primary field of research) reference style guide and do not utilize footnotes in this article. Notes included below each table are prepared in the style requested by Healthcare.
Looking at the work as a whole, it is well presented. The solution of the problem is well set out in the conclusion of the work.
AUTHOR RESPONSE: We appreciate your comments and support of the article’s presentation.
It is very good that the authors call for additional research, including a possible prioritization of scriptural resources and otherworldly sensibilities over scientific reason and this-worldly considerations. Bridging the gaps between faith and reason as competing perspectives in our current times is among the most pressing challenges faced by those concerned about public health. Though it is difficult to believe in our polarized times, faith and reason may be redefined as complementary facilitators of good health rather than competing worldviews.
There are no direct studies on this subject. It raises no objections in terms of content and methodology. The statistical methods have been applied correctly. The use of charts and tables allows readers to follow the author's scientific proceedings. The pattern of proceedings presented in this way has led to conclusions that are consistent with the evidence and reasoning applied. In my opinion the paper is of highest value and therefore I would suggest publishing it.
AUTHOR RESPONSE: We greatly appreciate your thorough review and recommendation to publish the article.
Reviewer 3 Report
This is an essay on an extremely important topic. I am going to let reviewers more in tune with quantitative methods discuss the use of statistics etc., and focus on what I found most interesting here. In spite of claims about the "otherworldly" outlook of conservative Protestants, they are overwhelmingly engaged in consumer culture, capitalism and activism. Indeed, when it comes to use of conventional Western medicine, they do not, on the whole, avoid hospitals, nursing homes and rehab centers. The "scepticism" and mistrust of science is selective. They want extreme medical treatment for their dying family members, when doctors tell them it is futile. "We expect a miracle" is one rationale for this- but behind this is the insistence that "Doctors aren't gods." The pandemic brought out the worst of this stance, fueled by some religious leaders and lots of politicians. Although 93% of American religious congregations observed lockdown restrictions during the pandemic, the small groups that did not were undoubtedly among those studied in this essay- or their spiritual siblings.
I would really like to see how the selection of some medical information and recommended health practices to be sceptical about works here. Do these same people practice artifical birth control? Diet for health reasons? Accept plastic surgery for cosmetic reasons? If so, what makes them un-sceptical here, but very mistrustful of other health recommendations? This lies outside the topic of the essay, but I believe generic "scepticism about science" is misleading if it is not taken into examination of specific channels of information, influence and practices.
Author Response
REVIEWER 3
This is an essay on an extremely important topic. I am going to let reviewers more in tune with quantitative methods discuss the use of statistics etc., and focus on what I found most interesting here. In spite of claims about the "otherworldly" outlook of conservative Protestants, they are overwhelmingly engaged in consumer culture, capitalism and activism. Indeed, when it comes to use of conventional Western medicine, they do not, on the whole, avoid hospitals, nursing homes and rehab centers. The "scepticism" and mistrust of science is selective. They want extreme medical treatment for their dying family members, when doctors tell them it is futile. "We expect a miracle" is one rationale for this- but behind this is the insistence that "Doctors aren't gods." The pandemic brought out the worst of this stance, fueled by some religious leaders and lots of politicians. Although 93% of American religious congregations observed lockdown restrictions during the pandemic, the small groups that did not were undoubtedly among those studied in this essay- or their spiritual siblings.
I would really like to see how the selection of some medical information and recommended health practices to be sceptical about works here. Do these same people practice artifical birth control? Diet for health reasons? Accept plastic surgery for cosmetic reasons? If so, what makes them un-sceptical here, but very mistrustful of other health recommendations? This lies outside the topic of the essay, but I believe generic "scepticism about science" is misleading if it is not taken into examination of specific channels of information, influence and practices.
AUTHOR RESPONSE: Thank you for your thoughtful questions and direct commentary. Our emphasis is on the relationship between COVID-19 pandemic response and conservative Protestantism in the selection of what behaviors are viewed as acceptable and not acceptable. We now include several references detailing aspects of religious pandemic responses not previously mentioned (e.g., political media portrayal versus public perspectives, instances of public health promotion by congregations, though these were often not conservative Protestant houses of worship). We have refined our discussion section to include a future pathway of comparative health behaviors between pandemic and non-pandemic perspectives of conservative Protestants. A mention of selective skepticism of scientific methods and applications has also been included.